# An Adapted GeneSwitch Toolkit for Comparable Cellular and Animal Models: A Proof of Concept in Modeling Charcot-Marie-Tooth Neuropathy

**DOI:** 10.3390/ijms242216138

**Published:** 2023-11-09

**Authors:** Laura Morant, Maria-Luise Petrovic-Erfurth, Albena Jordanova

**Affiliations:** 1Center for Molecular Neurology, VIB, University of Antwerp, 2610 Antwerpen, Belgium; laura.morant@uantwerpen.vib.be (L.M.); maria-luise.petrovic-erfurth@uantwerpen.vib.be (M.-L.P.-E.); 2Department of Biomedical Sciences, University of Antwerp, 2610 Antwerpen, Belgium; 3Molecular Medicine Center, Department of Medical Chemistry and Biochemistry, Faculty of Medicine, Medical University-Sofia, 1431 Sofia, Bulgaria

**Keywords:** disease modelling, *Drosophila melanogaster*, GeneSwitch™, aminoacyl tRNA-synthetase, Charcot–Marie–Tooth neuropathy

## Abstract

Investigating the impact of disease-causing mutations, their affected pathways, and/or potential therapeutic strategies using disease modeling often requires the generation of different in vivo and in cellulo models. To date, several approaches have been established to induce transgene expression in a controlled manner in different model systems. Several rounds of subcloning are, however, required, depending on the model organism used, thus bringing labor-intensive experiments into the technical approach and analysis comparison. The GeneSwitch™ technology is an adapted version of the classical UAS-GAL4 inducible system, allowing the spatial and temporal modulation of transgene expression. It consists of three components: a plasmid encoding for the chimeric regulatory pSwitch protein, Mifepristone as an inducer, and an inducible plasmid. While the pSwitch-containing first plasmid can be used both in vivo and in cellulo, the inducible second plasmid can only be used in cellulo. This requires a specific subcloning strategy of the inducible plasmid tailored to the model organism used. To avoid this step and unify gene expression in the transgenic models generated, we replaced the backbone vector with standard pUAS-attB plasmid for both plasmids containing either the chimeric GeneSwitch™ cDNA sequence or the transgene cDNA sequence. We optimized this adapted system to regulate transgene expression in several mammalian cell lines. Moreover, we took advantage of this new system to generate unified cellular and fruit fly models for YARS1-induced Charco–Marie–Tooth neuropathy (CMT). These new models displayed the expected CMT-like phenotypes. In the N2a neuroblastoma cells expressing YARS1 transgenes, we observed the typical “teardrop” distribution of the synthetase that was perturbed when expressing the YARS1^CMT^ mutation. In flies, the ubiquitous expression of YARS1^CMT^ induced dose-dependent developmental lethality and pan-neuronal expression caused locomotor deficit, while expression of the wild-type allele was harmless. Our proof-of-concept disease modeling studies support the efficacy of the adapted transgenesis system as a powerful tool allowing the design of studies with optimal data comparability.

## 1. Introduction

The systematic molecular investigation of protein function and signal transduction pathways often requires the parallel establishment of in cellulo and in vivo models allowing regulated expression of transgenic constructs (e.g., cDNAs, dsRNAs, shRNAs, and gRNAs). To achieve this, a transgene is usually delivered on expression plasmids to the experimental model system together with a suitable selection marker using different methods, such as transfection, transduction, electroporation or microinjections, etc. [1]. Traditionally, the plasmid backbone defining the promoter used, as well as the method of selection, is heavily dependent on the model organism and is also influenced by the preference of the investigator and the laboratory. The parallel investigation of transgenic effects within the same (such as cell lines) or distinct model system often involves subcloning. Thus, it not only requires time and reagents but it is also challenging with respect to the comparison of results established with different model organisms and within distinct laboratories/studies. This problem is especially relevant in the field of disease modeling, where investigators use different in cellulo and in vivo models to study the effect of disease-causing mutations in human genes to ascertain underlying molecular mechanisms and to test therapeutic approaches.

Based on the promoter choice, it is possible to achieve two distinct temporal modes of expression, namely the constitutive or the conditional expression of a transgene. While constitutive expression (e.g., using an actin or CMV promoter) guarantees uniform strong presence of the transgene, it is important to consider that such a strategy might create conditions far away from the physiological condition. This might lead to unwanted effects, such as unusual transgene localization, unwanted transgenic protein aggregation and protein–protein interactions, as well as in cellulo or in vivo toxicity [1]. This might hamper the in-depth understanding of the context-dependent regulation, the tissue specificity, and the molecular pathway(s) in which the transgene might be involved.

Several conditional gene expression systems have been developed over the years, enabling the researcher to control transgene expression in a spatial and temporal manner. Classically, an environmental stimulus is used to activate transgene expression at a given time-point. Typical inducers are, for example, a heat shock [2,3], the presence of an antibiotic (tetracycline or doxycycline) [4], (heavy) metal ions (cadmium [5,6], zinc [7], copper [8]), galactose and its analogs (e.g., isopropyl beta-d-thiogalactosidase (IPTG), thiomethyl galactosidase (TMG) [9]), or steroid hormones (estrogen [10], glucocorticoids [11,12]. More sophisticated inducible systems are the bipartite expression systems requiring the presence of two essential components at the same time and space to enable transgene expression, such as the UAS-GAL4 system [13], the Cre-LoxP system [14], the FLP-FRT-system [15], and the LexA system [16], as well as the most recently developed CRISPR-Cas9 system [17,18]. These bipartite systems enable very controlled spatial and temporal regulation of transgene expression.

For all inducible expression systems, care has to be taken to ensure that the chosen environmental cues do not display effects on the signal transduction pathway studied, they do not interfere with endogenous gene expression [19], and are not toxic to the system of interest [20,21,22,23,24,25]. In addition, many conditional promoters show basal levels of expression in the absence of an inducer (leakage) [26,27], rendering it necessary to evaluate the applicability of the expression system for each cell type/organism used. Historically, these individual considerations have led to the introduction of different induction systems that have been tested at different time-points for distinct model systems leading to variable popularity and traditions across different platforms (e.g., UAS-Gal4 in *Drosophila* (1990s), IPTG in *E. coli* (1960s) and Cre-LoxP in mice (1980s)).

Given that many labs nowadays use multiple cellular models for a particular study, an ideal inducible expression system should (1) be easily applicable to all cellular systems of interest, (2) require inducers that are non-toxic, (3) display low/no leakiness, (4) be already validated in as many cellular models as possible and (5) be controllable with high spatial and/or temporal resolution. One example complying with those criteria is the GeneSwitch™ technology, which is a conditional inducible system originally developed by Wang et al. [28] and commercialized by TermoFisher to be used exclusively in mammalian cells. It is an advanced version of the bipartite UAS-GAL4 regulatory framework containing the pSwitch regulatory vector, the Mifepristone as an inducer, and the pGene inducible plasmid. Classically, the yeast transcription activator protein GAL4 binds specifically to the Upstream Activation Sequence (UAS), an enhancer allowing gene transcription [29]. In the GeneSwitch™ system, the GAL4 cDNA sequence has been fused to the regulatory domain of the human progesterone receptor and the human p65 transactivator (Figure 1A). The resulting pSwitch regulatory fusion protein is expressed as a monomer under the control of a mild constitutive promoter. This monomer is, however, not capable of binding and activating a UAS-promoter, because its activation requires a GAL4-dimer (Figure 1B). Application of progesterone, or its analogue Mifepristone (RU486), induces a conformational change leading to dimerization of the pSwitch proteins (Figure 1C). This dimer is then capable of binding to the UAS sequence, allowing for transcriptional activation of any gene of interest placed under the control of this promoter (Figure 1D).

Here, we adapted the GeneSwitch™ technology to enable selection and conditional expression of a tagged transgene in two different model organisms, fruit fly and mammalian cells, without the need for subcloning. We established several mammalian stable cell lines carrying the novel regulatory vector to enable the expression of any inducible construct by means of transient transfection. The novel version of the inducible vector can be activated by GeneSwitch™/Mifepristone or GAL4 driver-containing plasmid, rendering it versatile to be used in both cell and fly models. For optimal transgene expression with minimal background signal detectable in the absence of an inducer (Mifepristone), we optimized the protocol for in cellulo induction based on the culture conditions. As a proof of concept, we generated and characterized in a standardized manner new cellular and fly models for YARS1-associated Charcot–Marie–Tooth neuropathy (CMT) which displayed established hallmarks of the disease [30,31], confirming the validity of the developed toolkit.


Figure 1Schematic illustration of the adapted pSwitch system. (**A**) Simplified scheme of the pSwitch regulatory cassette containing the yeast specific GAL4 DNA binding domain, the truncated human progesterone receptor ligand binding domain (hPR), and an activation domain from the human NF-KB p65 protein (p65). This cassette encodes for the (**B**) pSwitch regulatory fusion protein in a monomeric inactive state. (**C**) By adding Mifepristone (red), a human progesterone receptor antagonist, the pSwitch fusion protein undergoes a conformational change leading to homodimerization. (**D**) The homodimer binds to the yeast UAS sequence to activate transgene expression. We adapted the GeneSwitch™ system by generating (**E**) the inducible pSwitch-multi vector containing five UAS (5 × UAS), a T7 promoter, a TK minimal promoter, the pSwitch regulatory cassette (GAL4-hPR-p65), a simian virus 40 polyadenylation site (SV40), an attB sequence for site-directed insertion, an SV40 promoter, an EM7 promoter, an Hygromycin resistance gene, and the mini-white gene. (**F**) The inducible pUAS-attB-multi plasmid contains the same grey features as the pSwitch regulatory vector, the cDNA sequence of the transgene flanked by three tags (HA, V5 and Flag) and a Zeocin resistance gene. (**G**) Schematic representation of the experimental layout for which expression of the transgene in the pUAS-attB-multi vector can now be controlled in cells and flies using Mifepristone or GAL4 activation, respectively, to ensure optimal data comparability without the need for subcloning.
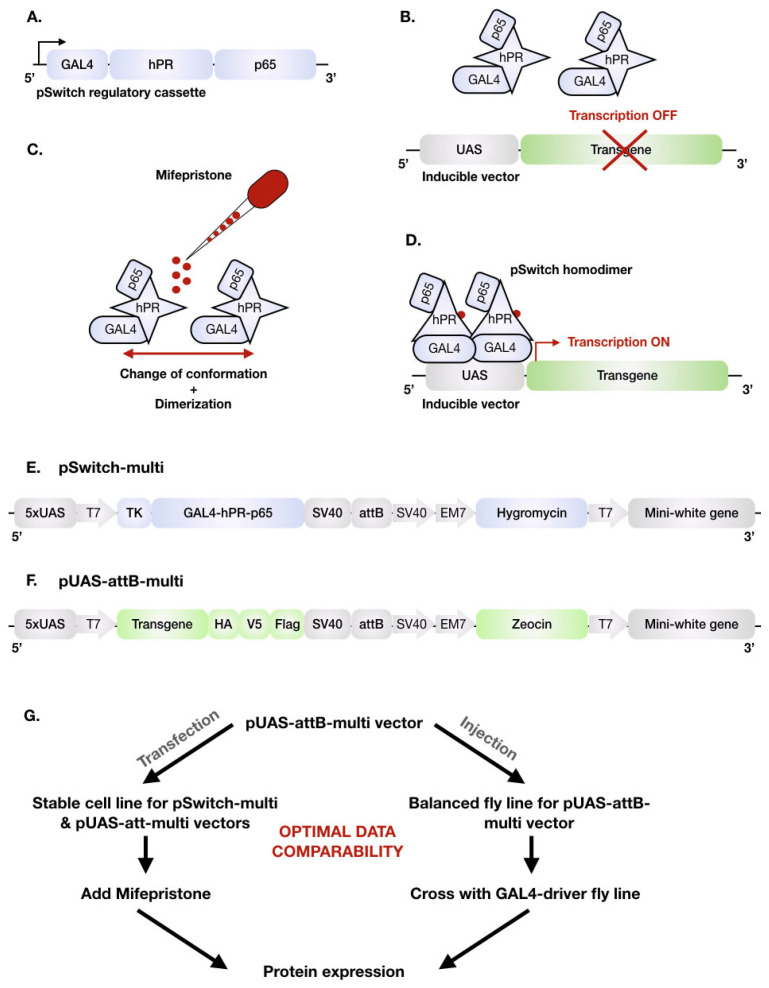



## 2. Results

In this study, we redesigned both the regulatory vector and the inducible plasmid of the GeneSwitch™ system so both plasmids can be selected in eukaryotic cells and injected into the fruit fly genome.

### 2.1. Adaptation of the GeneSwitch™ System to Avoid Subcloning

First, the pSwitch regulatory vector was redesigned as the pSwitch-multi regulatory vector (called the regulatory vector in the rest of the study). For this, we used the classical pUAST-attB as a backbone plasmid into which the GeneSwitch™ cassette (GAL4-hPR-p65) was inserted at the Multiple Cloning Site (MCS) (Figure 1A). This cassette contains the following features: (a) a Thymidine Kinase (TK) minimal promoter, (b) the yeast GAL4 DNA binding domain, (c) the *C*-terminal amino acids 640-914 of the human progesterone receptor ligand binding domain (hPR-LBD) and (d) the human NF-kB p65 activation domain (Figure 1A,E). We added five UAS motifs to allow a positive-feedback-loop expression when the system is activated. In addition, the Ampicillin-resistance gene was replaced by a Hygromycin-resistance gene to enable the antibiotic-based isolation of both prokaryotic and eukaryotic cells carrying the plasmid (Figure 1E). The vector contains an attB site allowing site-specific integration into the genome of the fruit fly (Figure 1E). Moreover, we inserted the mini-white gene coding sequence, which is a widely used selectable marker in fruit fly transformation constructs (Figure 1E).

Next, we generated an inducible pUAS-attB-multi vector to replace the pGene inducible vector from the GeneSwitch™ system (called inducible vector in the rest of the study) (Figure 1F). For this, the pUAS-attB plasmid served as a backbone that can harbor the coding sequence of any transgene of interest. In this study, we focused on two human aaRS genes, dominant mutations which cause clinical subtypes of CMT disease: tyrosyl-tRNA synthetase (*YARS1*) causing Dominant Intermediated CMT type C (DI-CMTC) and alanyl-tRNA synthetase (*AARS1*) causing CMT type 2 N (CMT2N). Their main transgene coding sequences were tagged with HA-V5-Flag triple tag at their C-terminus. We inserted five UAS sequences to induce transgene expression by (i) applying Mifepristone, together with the pSwitch-multi regulatory vector, or (ii) using the classical UAS-GAL4 system both in cellulo and in vivo. We also replaced the Ampicillin-resistance gene of the pUAST-attB vector with a Zeocin-resistance gene enabling antibiotic-based selection of prokaryotic and eukaryotic cells carrying the plasmid (Figure 1F). This construct also contains an attB site and the mini-white gene coding sequence.

Once the regulatory and the inducible plasmids are co-transfected in eukaryotic cells, the presence of a mild constitutive promoter in the regulatory vector allows the expression of the monomeric regulatory protein. The hPR-LBD domain is a fusion protein capable of binding synthetic progesterone antagonists such as Mifepristone, but not endogenous steroid hormones. This domain retains the regulatory protein in an inactive monomeric form by covering the GAL4 dimerization domain. Therefore, this monomer is not capable of binding and activating the UAS-promoter of the inducible vector as it requires a GAL4-dimer. By adding the recommended concentration of Mifepristone (10 nM), the regulatory protein undergoes a conformational change capable of dimerization. This homodimer then binds to the UAS-promoter present on the inducible vector to induce expression of the transgene of interest. It also binds to the UAS-promoter sequence of the regulatory vector, leading to a positive feedback loop in which the regulatory protein is overexpressed but not overactivated [28,32,33].

### 2.2. Promoting Versatility of the System by Inducing pUAST-attB-Multi Expression with pSwitch-Multi Regulatory VECTOR or Actin-GAL4 Driver Containing Vector

To streamline our in cellulo experiments, we generated several stable cell lines for the regulatory vector upon Hygromycin. These were the following: human cervical cancer (HeLa), human embryonic kidney 293 (HEK293T), mouse neuroblastoma (N2a), Chinese hamster ovary epithelial (CHO-K1), and *Drosophila* Schneider 2 (S2) cells. These stable lines can now be transiently transfected with any inducible plasmid for transgene expression using Mifepristone (10 nM). We tested the applicability of this system by transiently transfecting stable cell lines for the regulatory vector with 1 µg of an inducible plasmid carrying the *YARS1* cDNA sequence. We found that the recommended Mifepristone induction concentration (10 nM) was sufficient to induce robust transgene expression in cell lines derived from several eukaryotic cells (Figure 2, and for quantification please see Appendix A). While it performed in the desired way in N2a cells (Figure 2A), we observed exogenous YARS1 in the absence of Mifepristone in CHO-K1, HeLa and HEK293T cells (Figure 2B–D), suggesting an illegitimate (leaky) expression in non-induced cells. This side effect was also observed in non-mammalian S2 cells (Figure 2E) and was reported previously for the GeneSwitch™ system [33]. The leakage could be prevented in the novel system by reducing the amount of transiently transfected inducible plasmid to a concentration of less than 0.5 µg plasmid per transfection in HeLa cells (Appendix A).

In parallel, we evaluated whether the inducible plasmid could be activated in cellulo in the presence of a constitutively active driver, such as the ubiquitous Actin-GAL4 driver. For this, the inducible plasmid was co-transfected together with an Actin-GAL4 driver-containing plasmid in *Drosophila* S2 cells. After 24 h of transfection, YARS1 protein expression was detected in S2 cells (Figure 2F, Appendix A), confirming the leakage was due to the presence of the regulatory vector rather than the inducible plasmid. Overall, we demonstrated the versatility of the inducible plasmid, as it can be induced either by the regulatory vector together with Mifepristone or by a standard constitutively active driver (here, the Actin-GAL4 driver).

Strong expression of the transgene might create adverse effects such as toxicity, unusual transgene localization, protein aggregation and/or aberrant protein interactions. Therefore, we challenged our system to estimate the inducible plasmid concentration that was necessary to be transfected to reach YARS1 endogenous protein expression. To this end, an increasing concentration of inducible plasmid encoding for YARS1 protein was transiently transfected in N2a cells stably expressing the regulatory vector. As reported above, there was no detectable leakage in the N2a cells with 1 µg of transfected inducible plasmid and the YARS1 exogenous expression level reached the endogenous expression signal (Figure 3A). The same experiment was performed in HeLa cells where uncontrolled expression was displayed. In this cell line, it was necessary to decrease the concentration of the transfected plasmid to 0.5 µg to obtain negligible leakiness and have a similar protein expression to that of the YARS1 endogenous level (Appendix A).

Successful transfection may be affected by vector length, related to the size of the insert. Therefore, we assessed the effect of large cDNA transfection through alanyl tRNA-synthetase (*AARS1*, 106 KDa) in N2a and HeLa cells stably expressing the regulatory vector. Once more, we could reach the AARS1 endogenous expression level by transfecting 1 µg of inducible plasmid without detecting background leakage in non-induced N2a cells (Figure 3B), suggesting expression efficiency was not affected by the vector length in this cell line. Transient transfection with the same plasmid concentration in HeLa cells resulted in non-negligeable leakage that could be prevented by reducing the inducible plasmid concentration to 0.5 µg (Appendix A). However, 1 µg of plasmid transfection was required to reach endogenous protein expression (Appendix A). These results suggest that cDNA sequence size, and ultimately the inducible plasmid size, affects protein expression leakage in non-induced cells. Overall, the cell type and dependent leakage construct demonstrated that this adapted toolkit would require the optimization of transfected plasmid, with the quantity dependent on its use in cellulo.

### 2.3. Applicability of the System in New Models for YARS1-Associated Charcot–Marie–Tooth Neuropathy

So far, six dominant mutations in *YARS1* have been reported to cause DI-CMTC [31,34,35,36], a progressive lifelong disorder characterized by length-dependent axonal degeneration and demyelination of both motor and sensory neurons [34,35,36,37]. Five of the mutations are localized in the catalytic domain of the enzyme; however, a defect in aminoacylation activity is not a shared property among them [30,31]. To uncover the pathomechanism underlying YARS1^CMT^, we have characterized extensively three YARS1^CMT^ mutations both in cellulo and in vivo [30,31,38,39,40]. Taking advantage of the new GeneSwitch toolkit we developed, we set out to create in a standardized manner unified and novel DI-CMTC cellular and fly models, and compared their hallmark characteristics to the previously published ones.

To model the DI-CMTC mutations in neuronal cells, we used N2a cells where no leakage of exogenous YARS1 protein expression in non-induced conditions was observed. N2a stable cell lines for the regulatory vector were transiently transfected with YARS1^WT^- or YARS^E196K^-expressing constructs and the subcellular localization of YARS1 was evaluated after three days of differentiation. As reported in our previous study [31], both endogenous YARS1 and transgenic YARS1^WT^ concentrated in granular structures at the branching points and growth cones of the projecting neurites. As expected, overexpression of YARS1^E196K^ significantly reduced this specific localization pattern (45% ± 0.5%) and the mutant protein was more homogenously distributed throughout the differentiating cells, as compared to cells expressing YARS1^WT^ (96% ± 0.4%) (Figure 4A). Western blot analysis of protein extract obtained from N2a cells transfected with YARS1^WT^ and YARS1^E196K^ showed comparable expression of exogenous as well as endogenous YARS1 (Appendix A).

In parallel, we generated new fly models for DI-CMTC using the attB site within the inducible vector to mediate site-specific transgenesis in the fruit fly genome using PhiC31 integrase. We used the attp40 recombination site on the *Drosophila*’s second chromosome and 86Fb recombination site on the third chromosome, allowing biological comparison. With this approach, we avoided position effects of the transgene compared to the conventional random transgenesis. Moreover, it enabled us to better compare the effect of the CMT mutation, as exogenous protein expression levels are expected to be consistent over different fly lines. We evaluated the effect of YARS1^WT^ and YARS1^E196K^ transgene ubiquitous and pan-neuronal overexpression (Actin-GAL4 and nSyb-GAL4 drivers, respectively). Strong YARS1^E196K^ ubiquitous expression induced developmental lethality (~30% ecloding flies), while YARS1^WT^ had no effect on the fly eclosion ratio (Figure 4B and Appendix A). This phenotype could be partially reverted by reducing the transgene expression levels using a weaker Actin5c-GAL4 driver (~42% ecloding flies) (Figure 4B). In line with the transgene expression levels, the phenotype was more pronounced in the previously published YARS1^E196K^ fly model (no ecloding flies), where the YARS1^E196K^ protein expression was stronger compared to the new model (Figure 4C). Then, we assessed the locomotor performance of the transgenic flies in an automated negative geotaxis climbing assay. Pan-neuronal expression of YARS1^E196K^ in both published and new models induced locomotor performance defects (average of ~5.293 s for the first fly to reach the upper limit), in contrast to YARS^WT^-expressing flies (average of ~4.252 s for the first fly to reach the upper limit) (Figure 4D and Appendix A) [30]. Altogether, the newly generated uniformed DI-CMTC cellular and fly models recapitulated previously described hallmark phenotypes, supporting the validity of our transgenesis toolkit and opening opportunities for genotype–phenotype correlations across different model systems.

## 3. Discussion

The GeneSwitch™ system is an inducible bipartite system initially developed to conditionally induce transgene expression in mammalian cells. It is activated upon administration of an exogenous drug compound (Mifepristone), and it can be easily terminated upon its withdrawal [28,32]. The drug is non-toxic and does not activate endogenous genes [28]. The bipartite nature of this system allows for excellent control of spatial and temporal expression. Nonetheless, it requires the subcloning of the plasmid carrying the transgene of interest in different inducible vectors, depending on the model organism used.

In this study, we adapted this technology by integrating the GeneSwitch™ cassette into a pUAS-attB vector. Moreover, we adjusted the pUAS-attB vector to a pUAS-attB-multi-inducible vector to generate versatile models (in this case, stable cell lines and transgenic flies) without the requirement of subcloning (Figure 1). Redesigning the GeneSwitch™ tool now offers the possibility of increasing the data comparison between these two model systems (Figure 1G). Thus, any other transgene can effectively be cloned into the inducible plasmid’s multiple cloning site, using overlapping primers. Expression of the transgene can be induced either by (i) Mifepristone in presence of the regulatory vector or (ii) by any GAL4-driver-containing vector (Figure 1C). The existence of different selection cassettes in both constructs allows the establishment of stable cell lines for both plasmids.

To investigate protein function and pathway regulation, transient transfection of the transgene of interest is routinely used. Because our adapted GeneSwitch system contains two plasmids, it requires a co-transfection strategy that might impair transfection efficiency, cell viability, and data interpretation, due to co-transfection variability (reviewed in [41]). Therefore, we established several stable cell lines for the pSwitch-multi-regulatory vector allowing a straightforward transient transfection or generation of stable cell lines of any pUAS-attB-multi-inducible vector. This is particularly relevant in the context of disease modelling or gene function analysis in which several variants are modelled for a systematic comparative molecular investigation. Using our modified toolkit, we established that transgene expression could be triggered in cellulo upon pSwitch–multi-Mifepristone activation or constitutive Actin-GAL4-containing plasmid, thus providing several strategies for transgene expression.

Apart from generating cells lines, we established new fly models for DI-CMTC using site-specific transgenesis for the pUAS-attB-multi-YARS1 inducible construct. To characterize these new fly models, protein expression was driven using the classical UAS-GAL4 technology. Nonetheless, the pUAS-attB-multi-inducible vector could potentially be activated in vivo using GeneSwitch™ for conditional spatial- and tissue-specific expression, as demonstrated during our in cellulo work. In fact, the Bloomington Drosophila Stock Center offers a variety of regulatory GAL4 drivers activated upon Mifepristone administration [20,42,43,44,45,46,47,48,49]. The use of a Mifepristone-dependent Actin driver might be particularly relevant for overcoming the developmental lethality observed in our standard Actin-GAL4 > YARS1^CMT^ fly model (Figure 4A). Using this system would allow us to evaluate the behavioral and molecular consequences of YARS1^CMT^ ubiquitous expression at the adult stage by administrating Mifepristone post eclosion.

Mifepristone is a clinically approved drug used for emergency contraception with an excellent safety profile [50], and its administration in rats is not abortifacient at low doses (<0.5 µg/kg) [51]. Its use in vivo requires a low concentration (<33 µg/mL) that does not affect fruit fly viability, development, or lifespan [20,48,52]. We demonstrated that the recommended concentration of Mifepristone (10 nM) was still sufficient to induce strong transgene expression using our newly adapted system (Figure 2), encouraging once more Mifepristone usage in vivo. Yet, precaution must be taken regarding the choice of the GAL4 driver in combination with the concentration of Mifepristone used. Indeed, a range of developmental abnormalities associated with the use of Tubulin-GS driver has been identified in combination with Mifepristone concentrations above those commonly used [53]. Therefore, users of the GeneSwitch™ system in vivo should consider the potential secondary effects of its application in particular circumstances.

Alternatively, a toolkit combining the CRISPR-Cas9 and the GeneSwitch™ technology has been developed, as constitutive Cas9 overexpression induces significant lethality in flies independently of its endonuclease activity [54,55]. Using tissue-specific gRNA resulted in a viable and powerful strategy compared to the common GAL4-mediated Cas9 overexpression [55]. Thus, functional studies of a protein could also be considered by using this toolkit for specific tissue or subcellular expression.

One limitation of the GeneSwitch™ system is the basal protein expression in the absence of the inducer (Mifepristone) [42,46,47]. This leakage seemed to be transgene (with less leakage when AlaRS was overexpressed compared to TyrRS) and cell-type dependent (in HEK293T cells and not N2a cells). Yet, the latter issue could be prevented by reducing the amount of transiently transfected plasmid.

The newly generated toolkit offers major advantages to both the cell and fly communities, as it will facilitate the translation from one model to another. In this study, we focused on YARS1, a ubiquitously expressed enzyme causing specific peripheral nerve degeneration upon dominant mutations. We applied our toolkit to generate new cellular and fly models for DI-CMTC. Both types of models displayed phenotypes that have been reported previously by our research group [30], such as YARS1 protein localization defects in cellulo and dosage-dependent developmental lethality and locomotor performance deficits in vivo (Figure 4A). These data demonstrate that our adapted GeneSwitch™ system is a robust tool that can be used to generate unified cellular and fly models to investigate pathogenic molecular mechanisms induced by disease-causing gene mutations. This toolkit will provide experimental consistency as well as reproducibility across these two model systems. Incorporating a novel transgene into pre-existing plasmid requires minimal effort, as there is no need to subclone for cells and flies. Moreover, a large number of strains already exist for the classical UAS-GAL4 and the Mifepristone/UAS-GAL4.

In the future, it would be interesting to extend this adapted toolkit to other model organisms such as mice, rat, zebrafish, or worms. As a matter of fact, the GeneSwitch™ technology was previously tested to prevent a lethal knockout phenotype in inhibin α null mice [56]. Its application was also reported in a humanized rat model to test controlled GDNF expression as a treatment strategy for Parkinson’s disease [57]. To our knowledge, the GeneSwitch™ was not tested in model organisms such as zebrafish and worms. Yet, the application of the classical UAS-GAL4 system was successfully reported in these animals [58,59], offering the opportunity to use either the adapted GeneSwitch™ toolkit or only the pUAS-attB-multi inducible construct.

The impact of the GeneSwitch™-based disease modeling for YARS1 is even greater when considering the other aminoacyl-tRNA synthetases implicated in CMT. Indeed, dominant mutations in six other aminoacyl tRNA-synthetases (*AARS1* [60], *GARS1* [37], *HARS1* [61], *MARS1* [62], *SARS1* [63], and *WARS1* [64]) have been associated with inherited peripheral neuropathies. Aminoacyl tRNA-synthetases (aaRS) are ubiquitously expressed enzymes catalyzing the aminoacylation reaction by covalently linking an amino acid to their cognate tRNA [65]. Loss of function does not induce CMT, suggesting a neurotoxic gain in function mechanism [30,31,66,67,68,69,70]. In fact, specific CMT causing aaRS mutations have been extensively studied in various in vitro and in vivo models. The combined data strongly supports the hypothesis that dominant aaRS-mutations induce CMT by a common molecular mechanism distinct from aminoacylation. Such commonalities have been reported at the genetic [40], structural [71,72,73] and functional level [74,75,76] (reviewed in [77,78]). However, the exact nature of this common CMT-causing toxic function remains unknown and a source of speculation, thereby limiting the development of an efficient therapeutic strategy. The problem is aggravated by the many different models in use, creating difficulties in comparing the data obtained in distinct studies. By establishing with our toolkit such a unified platform for aaRS^CMT^, we will not only reveal common pathogenic mechanisms in CMT-causing aaRS mutations, but we will also facilitate the design of an effective common therapeutic strategy for this pathology.

## 4. Materials and Methods

### 4.1. Cloning and Plasmids

All expressing vectors were generated using Gibson Assembly^®^, according to manufacturer’s recommendations (New England Biolabs, Ipswich, MA, USA). The pUAST-attB cDNA, including a *C*-terminal HA-Flag-V5 tag, was amplified from a previously reported construct [79]. PCR primers and Gibson-overlap arms are summarized in Table 1. The Hygromycin and Zeocin selection cassettes were PCR-amplified from the pLenti vector and subsequently cloned into the pUAST-attB vector.

The first part of the regulatory cassette containing GAL4-hPR was generated by the DNA synthesis Service Core at VIB (Vlaams Instituut voor Biotechnologie, Flanders, Belgium). The second part of the regulatory cassette consisting of the p65 was amplified from pGEM-RELA plasmid (Sino Biological Inc., Beijing, China) The regulatory cassette was subsequently cloned into the pUAST-attB vector containing the Hygromycin resistance cassette to create the pSwitch regulatory vector (Table 1). Full-length human *YARS1* cDNA was amplified and subsequently cloned into the pUAST-attB vector containing the Zeocin resistance cassette to create the inducible vector (Table 1). All constructs were transformed into chemically competent EPI300 bacteria. The selection of bacteria containing the construct of interest was performed by using Hygromycin-B (50 µg/mL, Invivogen, San Diego, CA, USA) or Zeocin (25 µg/mL, Thermo Fisher Scientific, Waltham, MA, USA). Plasmids were purified using Nucleospin Plasmid easy pure kit™ (Macherey-Nagel, Düren, Germany). All constructs were Sanger-sequencing verified.

### 4.2. Cell Culture

The HEK293T (human embryonic kidney) cell line was purchased from American Type Culture Collection (ATCC, #CRL-3216, #CRL-2266, Manassas, VA, USA). The S2-DGRC (*Drosophila* embryonic Schneider) cell line was purchased from the Drosophila Genomics Resource Center (DGRC, Bloomington, IN, USA). The HeLa (human cervical cancer) cell line was purchased from DSMZ German collection of Microorganisms and Cell Culture (ACC 57, Leibniz Institute, Germany). The N2a (mouse neuroblastoma) and CHO-K1 (Chinese hamster ovary) cell lines were a gift from Prof. Vincent Timmerman (University of Antwerp, Belgium).

Mammalian cell lines were grown at 37 °C in a humidified atmosphere containing 5% CO_2_ in DMEM-F12 (Gibco, Waltham, MA, USA) supplemented with 10% heat-inactivated fetal bovine serum (FBS, Gibco, Waltham, MA, USA), 1% Penicillin-Streptomycin (P/S, Gibco, Waltham, MA, USA) and 1% non-essential amino-acids (Gibco, Waltham, MA, USA). The S2-DGRC cell line was grown at 25 °C in Schneider’s Insect medium (Gibco, Waltham, MA, USA) supplemented with 15% FBS (Gibco, Waltham, MA, USA) and 1% P/S (Gibco, Waltham, MA, USA). N2a cells were differentiated by incubation in complete media without FBS, for 72 h.

### 4.3. Transfection and Stable Cell Line Generation

CHO-K1, HEK293T, HeLa and N2a cell lines were seeded and transfected using polyethylemnimine (PEI) transfection reagent (Thermo Fisher Scientific, Waltham, MA, USA) (Table 2). To generate stable cell lines, cells were seeded out in 6-well plates (Table 1). On the day of the transfection, 1 µg of plasmid DNA was diluted in 115.4 µL Opti-MEM (Life Technologies, Waltham, MA, USA) and, in parallel, 7.2 µL PEI (1 µg/µL) was diluted in 115.4 µL Opti-MEM. The diluted PEI was then added to the diluted DNA and mixed by vortexing for 10 s. The PEI-DNA mix was incubated for 10 min at room temperature, and then added dropwise onto the cells.

The S2-DGRC and N2a cell lines were transfected using Genius DNA Transfection reagent (Westburg, Grubbenvorst, The Netherlands) (Table 1). Briefly, cells were seeded out in 6-well plates (Table 2). On the day of transfection, 4 µg of plasmid DNA was diluted in 400 µL Opti-MEM. Afterwards, 6 µL of GENIUS reagent was added to the diluted DNA and mixed by vortexing for 10 s. The mix was incubated at room temperature for 15 min, and then added dropwise onto the cells. Hygromycin antibiotics were used 24 h post transfection at different concentrations, depending on the cell line, to generate stable cell lines for the pSwitch regulatory vector (Table 2). Stable cell lines for the pSwitch-multi regulatory vector were transiently transfected using either PEI or Genius transfection reagent, depending on the cellular model (Table 2). Four hours after transfection, the medium was refreshed, and protein expression was induced using the concentration of Mifepristone (10 nM) recommended by the manufacturer (Thermo Fisher Scientific, Waltham, MA, USA). Cells were harvested 24 h to 72 h post induction for protein extraction.

### 4.4. Western Blot

Cell lines were lysed in radioimmunoprecipitation assay buffer (RIPA—50 mM Tris pH 7.6, 150 mM NaCl, 0.5% Sodium Deoxycholate, 1% NP40, 0.1% SDS) supplemented with a protease inhibitor cocktail (1X, Sigma Aldrich, St. Louis, MO, USA). Lysates were prepared at a concentration of 1 µg/µL in 250 µM DTT (Merck, St. Louis, MO, USA) in Laemmli buffer (Biorad, Hercules, CA, USA). Samples were boiled at 95 °C for 10 min before being loaded (20 µg) on 10% Mini-PROTEAN TGX™ PAGE Precast gels (Biorad, Hercules, CA, USA) and run for 1 h at 90 V. Protein transfer was performed on Membrane Protean Premium 0.45µm pore size nitrocellulose (VWR, Radnor, PA, USA) using standard preprogrammed protocol (30 min, 1 A up to 25 V) from the Transfer-Blot^®^ Turbo™ Transfer System (Biorad, Hercules, CA, USA). The following antibodies were used in different Western blot experiments: mouse monoclonal HA-tag (1:1000, 2367S, Cell Signaling Technology, Beverly, MA, USA), mouse monoclonal YARS1 (1:1000, H00008565-M02A, Abnova, Taipei, Taiwan), mouse monoclonal AARS1 (1:1000, sc-165991, SantaCruz Biotechnologies, Dallas, TX, USA), mouse monoclonal α-tubulin (1:5000, ab7291, Abcam, Cambridge, UK), goat anti-mouse IgG_1_ (1:10,000, 1070-05, Sanbio B.V., Uden, The Netherlands), and goat anti-mouse IgG2a (1:10,000, 1080-05, Sanbio B.V., Uden, The Netherlands). Western blot signal was detected using chemifluorescent detection with Pierce ECL Plus™ (Thermo Fisher Scientific, Waltham, MA, USA) on Amersham Imager 600™ (GE-Healthcare, Chicago, IL, USA). The amount of protein was semi-quantitatively determined based on signal density of the area, using standard tools of the ImageJ software (v1.53s).

### 4.5. Immunohistochemistry

N2a cells were fixed three days after differentiation, using 4% paraformaldehyde. Staining procedure involved 2 washes in PBS, permeabilization using 0.1% Triton X100 and staining with mouse monoclonal HA-tag (1:1000, 2367S, Cell Signaling Technology, Beverly, MA, USA), mouse monoclonal YARS1 (1:500, H00008565-M02A, Abnova, Taipei, Taiwan), Alexa Fluor^®^ 488 goat anti-mouse IgG (HL) antibody (1:250, A11001, (Thermo Fisher Scientific, Waltham, MA, USA), and AlexaFluor™ 594 Phalloidin (Thermo Fisher Scientific, Waltham, MA, USA), according to the manufacturer’s instructions. Cells were imaged using a confocal laser scanning microscope Carl Zeiss LSM700 equipped with a 40× (1.3 NA) Plan-Neofluar objective. Voxel size: 0.1563 × 0.1563 × 0.9474 µm^3^.

### 4.6. Fly Genetics

Transgenic flies were generated by Rainbow Transgenic Flies, Inc. (Camarillo, CA, USA) using phiC31 integrase combined fly lines. The following insertion sites were used to integrate pUAST-attB-multi-YARS1 into the fruit fly genome: attp40 recombination site (P{nos-phiC31\int.NLS}X, P{CaryP}attp40, 25709, RTF) for the 2nd chromosome insertion and 86Fb recombination site (M{vas-int.Dm}ZH-2A, M{3xP3-RFP.attP}ZH-86Fb 24749, RTF) for the 3rd chromosome insertion. Transgenic fly lines were then balanced on CyO for the 2nd chromosome insertion and TM3, Sb for the 3rd chromosome insertion. The *Act5C-GAL4^strong^* and *Act5C-GAL4^weak^* driver lines correspond to *P{Act5C-GAL4}25FO1* and *P{Act5C-GAL4}17bFO1* lines from Bloomington Drosophila Stock Center (BDSC). All crosses were performed at 25 °C, 12 h light/dark cycle, on standard NutriFly medium (FlyStuff Genesee Scientific, El Cajon, CA, USA).

### 4.7. DNA Isolation and Sequencing

We proceeded to single-fly genomic DNA preparation using Squishing buffer (10 mM Tris-HCl pH = 8, 1 mM EDTA, 25 mM NaCl, 200 μg/mL proteinase K). Samples were incubated at 37 °C for 30 min and heated to 95 °C for 3 min to inactivate the Proteinase K. After spinning down the samples briefly, the supernatant was transferred to 500 μL ethanol. Centrifugation of 1 min at 140,000 rpm was performed to precipitate the DNA that was resuspended in elution buffer. The concentration and purity of the isolated DNA was assessed using Nanodrop (Thermo Fisher Scientific, Waltham, MA, USA) measurement. The insertion sites were amplified by RT-PCR using TiTaq polymerase (Takara Bio, Kyoto, Japan) paired with primers specific to the YARS1 sequence or to sequences on the 2nd or 3rd chromosome insertion (Table 3). For each construct, multiple transgenic fly lines were established and the sequence verified using Sanger sequencing (Table 3).

### 4.8. Protein Expression Level in Flies

Expression level of the transgene was induced by Act5C-GAL4^weak^ and determined using Western blot for each fly line. Briefly, 10 fruit flies per genotype were collected and lysed in 100 μL RIPA buffer and Western blot was performed as described above.

### 4.9. Lethality Assay

Both Act5C-GAL4^weak^/TM3 and *Act5C-GAL4^strong^/CyO,GFP* were used to perform the developmental lethality assay [38,40]. Briefly, adult *Drosophila* eclosion ratios were determined by manually counting the number of *Act5C-GAL4* > transgene versus the number Balancer (TM3 or CyO, GFP) > transgene adult flies.

### 4.10. Negative Geotaxis Assay

To assess the fruit fly’s motor performance, a negative geotaxis assay [80] was performed on adult age-matched female flies (nSyb-GAL4 > transgene) using the semi-automated FlyCrawler device (Peira Scientific Instrument, Beerse, Belgium), as previously described [38]. Adult female flies with shortened wings were kept until day 10 after eclosion and placed in a cylindrical vial (49 mm). An infrared camera allowed us to record the fly crawling on a vertical wall from the moment they were shaken down to the bottom of the vial until they reached the height of 82 mm. For each genotype and condition, 10 groups of 10 flies were tested 15 times, and the average of the 10 fastest flies was calculated.

## Figures and Tables

**Figure 2 ijms-24-16138-f002:**
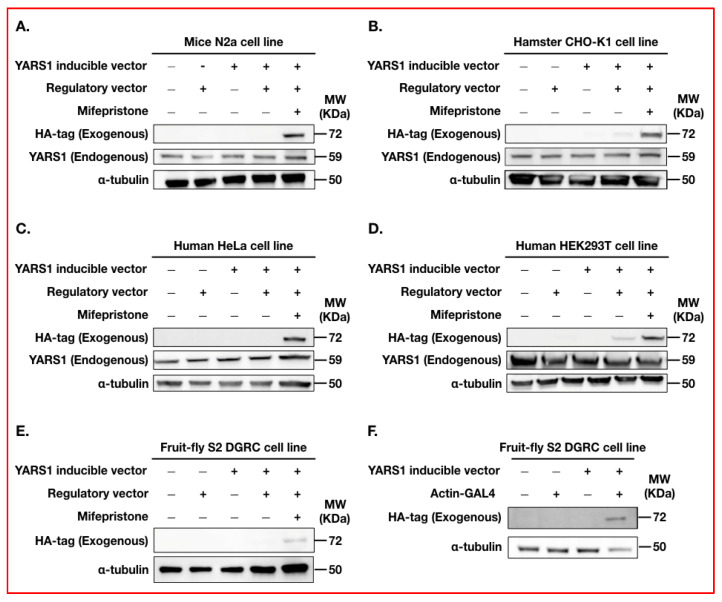
Induction of YARS1 expression using the pSwitch-multi system in mammalian and invertebrate cell lines. Western blot analysis of protein extracts from (**A**) N2a, (**B**) CHO-K1, (**C**) HeLa, (**D**) HEK293T and (**E**) S2 stable and non-stable cell lines that have been transiently transfected with 1 µg of pUAST-attB-multi-YARS1. Four hours after transfection, a recommended concentration of Mifepristone (10 nM) was added to the medium in the conditions containing the two plasmids, for 3 days. (**F**) Western blot analysis of protein extract from S2 cells 24 h after transient co-transfection with pUAST-attB-multi-YARS1 and/or Actin-GAL4 driver (1:1 transfection ratio). Expression of exogenous YARS1 was detected using mouse monoclonal HA-tag antibody. Endogenous YARS1 expression could be detected in mammalian cells by using mouse monoclonal YARS1 antibody. Equal loading was validated by using mouse monoclonal α-tubulin antibody (*n* = 3). Note the exogenous YARS1 signal detection in the condition containing both vectors but without mifepristone induction in HeLa, CHO-K1 and HEK293T cells.

**Figure 3 ijms-24-16138-f003:**
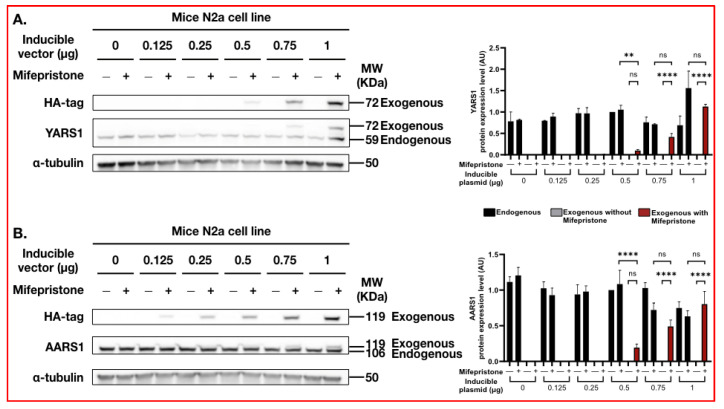
Concentration of inducible plasmid to be transfected in stable cell lines for pSwitch-multi regulatory vector. N2a cells stably expressing the pSwitch-multi regulatory vector have been transiently transfected with increased concentrations of inducible plasmid (ranging from 0 to 1 µg) containing (**A**) YARS1 or (**B**) AARS1 cDNA sequence. Four hours after transfection, a recommended Mifepristone concentration (10 nM) was added to the medium to induce transgene expression for 24 h. The transgene signal was first determined by using the HA-tag antibody. Exogenous and endogenous YARS1 and AARS1 expression was detected with mouse monoclonal YARS1 or AARS1 antibodies, respectively. Equal loading was validated by using mouse monoclonal α-tubulin antibody. Each graph represents the relative quantification of each band’s intensity compared to the expression level at 0.5 µg inducible plasmid without the presence of Mifepristone (*n* = 3). Bar charts are represented with s.e.m. for endogenous (black), exogenous in absence (grey) or presence of Mifepristone (red) YARS1 and AARS1, respectively. Statistical significance (** *p* < 0.01, **** *p* < 0.0001, ns—not significant) was determined using one-way ANOVA analysis.

**Figure 4 ijms-24-16138-f004:**
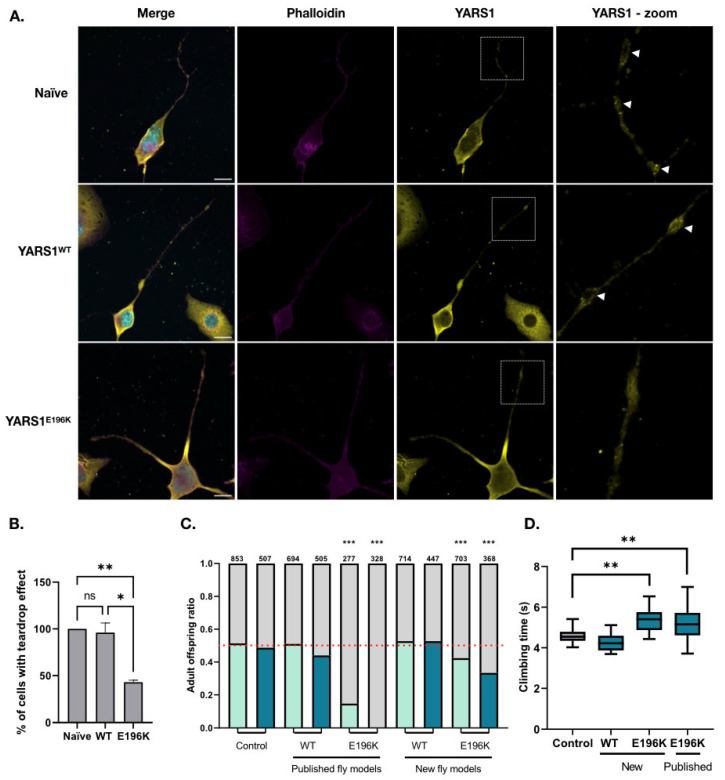
Characterization of unified cellular and fly models for YARS^CMT^. (**A**) Immunofluorescence analysis of differentiated N2a cells transfected with wild-type and mutant YARS1 inducible plasmids. YARS1—yellow, nuclei—cyan, actin cytoskeleton—magenta (Scale bar = 20 µM). A zoom-in on the YARS1 staining is represented by a white box. The teardrop effect is indicated by a white arrowhead. (**B**) Quantification of cells displaying the teardrop effect. Statistical significance ** *p* < 0.01, * *p* < 0.1, ns—not significant was determined after one-way ANOVA analysis (*n* = 4). (**C**) Strong ubiquitous expression of YARS1^WT^ has no effect with Actin5c-GAL4weak (light blue) and Actin5c-GAL4strong (dark blue) on the expected 1:1 adult flies eclosion ratio. YARS1^E196K^ ubiquitous expression has detrimental effects in a dosage-dependent manner in both new and previously published models. Attp40 flies were used as a negative control. The number of adult flies ecloding is indicated above each graph bar. Dashed line marks the expected 1:1 genotypes’ eclosion ratio. Statistical significance (*** *p* < 0.0001) was determined after One-Way ANOVA analysis which compares the odd ratios (Actin5c-GAL4/transgene ON vs. Balancer/transgene OFF) of flies in three independent crosses. (**D**) Pan-neuronal expression of mutant YARS1 with nSyb-GAL4 induces locomotor performance deficits as determined in a negative geotaxis climbing assay. NSyb-GAL4 flies were used as a negative control (grey). The Y-axis indicates the time needed for the fastest fly to climb a vertical wall to a height of 82 mm. ** *p* < 0.01 was determined after one-way ANOVA analysis (*n* = 3). Both lethality and negative geotaxis assays were performed on flies carrying the transgenic construct on the second chromosome.

**Table 1 ijms-24-16138-t001:** List of primers for creating the pSwitch-multi and the pUAST-attB-multi vectors. Primers in blue contain overlap arms with the vector and the different cassette to be inserted for the Gibson assembly. Lower case: sequence aligning with the cassette to be inserted. Upper case: sequence aligning with the host vector.

Sequence Amplified	Primers	Sens	Tm (°C)
pUAST-attB for new resistance cassette insertion	CATTTCCCCGAAAAGTGCCAC	Forward	60
	GGATCTAGGTGAAGATCCTTTTTGATAATCT	Reverse	
Hygromycin resistance	AGATTATCAAAAAGGATCTTCACCTAGATCCgtgtgtcagttagggtgtgga	Forward	60
	TCAGGTGGCACTTTTCGGGGAAATGctattcctttgccctcgg	Reverse	
pUAST-attB for GAL4-hPR-p65 insertion	cctgctgagtcagatcagctcctAACGGCCGCGACTCTAGATCATA	Forward	60
	CGCTAGAGTCTCCGCTCGG	Reverse	
GAL4-hPR	CCGAGCGGAGACTCTAGCGatgtcgaccccgccca	Forward	60
	aatgcgtcgaggtggag	Reverse	
P65	CTCCACCTCGACGCATTgctgtgccttcccgc	Forward	55
	TGATTATGATCTAGAGTCGCGGCCGttaggagctgatctgactcagcagg	Reverse	
pUAST-attB for transgene insertion	GCTGGCAAGCCCATCCC	Forward	65
	GGCTCCGGTACCCTCGA	Reverse	
Zeocin resistance	AGATTATCAAAAAGGATCTTCACCTAGATCCAatgagtttggaattaattctgtggaatgtgt	Forward	65
	GTGGCACTTTTCGGGGAAATGccccccttttcttttaaaaagtggc	Reverse	

**Table 2 ijms-24-16138-t002:** Condition for stable and transient transfection.

Cell Line	Cells per Well in 6-Well Plate	Transfection Reagent	Hygromycin Concentration	Amount of Plasmid Recommended for Transient Transfection
CHO-K1	2 × 10^5^/mL	PEI	800 µg/mL	Not tested
HEK293T	2 × 10^5^/mL	PEI	200 µg/mL	Not tested
HeLa	2 × 10^5^/mL	PEI	200 µg/mL	0.5 µg
N2a	3,5 × 10^5^/mL	Genius	200 µg/mL	1 µg
S2-DGRC	2 × 10^6^/mL	Genius	200 µg/mL	Not tested

**Table 3 ijms-24-16138-t003:** Primers used to sequence transgenic flies.

Sequence	Primers	Sens
Attp40 5′ site	TGCGTATACTCCCGTTTTGA	Forward
	CAGGAAACAGCTATGAC	Reverse
86Fb 5′ site	GGGTGCATGTGACCGTAAAT	Forward
	CAGGAAACAGCTATGAC	Reverse
Attp40 3′ site	AAGGGCATCGACTTCAAGGA	Forward
	AAGTCGCGAGAGAAGAGCTG	Reverse
86Fb 3′ site	AAGGGCATCGACTTCAAGGA	Forward
	GAACTACTGACTCAAACATGCAAT	Reverse
YARS1	TCGAGGGTACCGGAGCC	Forward
	ATCAATCTTGGACTCCTCTTCTGAA	Reverse

## Data Availability

All results obtained in this study are presented in this manuscript.

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
