# Peer review of "An Adapted GeneSwitch Toolkit for Comparable Cellular and Animal Models: A Proof of Concept in Modeling Charcot-Marie-Tooth Neuropathy"

_ijms, 2023, doi:10.3390/ijms242216138_

Round 1
Reviewer 1 Report
Comments and Suggestions for Authors
The authors describe a modification of an established technique for its use in more than one model, and it is especially suitable for disease models. The design, testing and validation of the novel constructs is rigorous, and therefore the results are robust. There is also a determination of possible drawbacks, mainly leaky expression from the vectors.
Minor comments.
In lines 67-70 the authors mention four different two-component systems, but they can be used for different goals (expression, editing). They should also mention the LexA system.
In general, the manuscript needs some spell checking (attp40 TermoFisher)
Comments on the Quality of English LanguageThe manuscript needs spell checking.
Reviewer 2 Report
Comments and Suggestions for Authors
The manuscript by Morant and colleagues describes an adaptation of the GeneSwitchTM technology to create a new regulatory vector. This vector allows the expression of any inducible construct in both cell and fly models using either GeneSwitch/Mifepristone or GAL4. They also created and characterized new cellular and fly models for YARS1-associated Charcot-Marie-Tooth neuropathy (CMT). These models displayed established hallmarks of the disease, confirming the validity of the developed toolkit.
Minor points:
1. Row 191: authors state “This side effect was also observed in non-mammalian S2 cells (Figure 2E). However, in figure 2E any band relative to exogenous YARS1 is observable in the absence of Mifepristone. Therefore, from this WB is difficult to claim that, also in S2, there is cell leakage due to the presence of the regulatory vector rather than the inducible plasmid (row 200). A WB image representing this result must be showed. Alternatively, the statement needs to be reformulated.
2. Ideally, all Western blots should be quantified. This is crucial, especially when the authors comment on the impact of inducible plasmid concentration on YARS1 and AARS levels. Without appropriate quantification relative to a housekeeping gene, it becomes difficult to assess the variation in expression of the exogenous proteins. Quantification would be necessary for all bands where leakage is observed (Figure 1A-F and Figure S1A and B), as well as for all bands relative to the various concentrations of the inducible plasmid's observed effects (Figure 3A and B).
3. It would be much more informative for the readers if the abstract primarily emphasizes the results obtained in the cellular and fly models.
